# B4GALT1 Is a New Candidate to Maintain the Stemness of Lung Cancer Stem Cells

**DOI:** 10.3390/jcm8111928

**Published:** 2019-11-09

**Authors:** Claudia De Vitis, Giacomo Corleone, Valentina Salvati, Francesca Ascenzi, Matteo Pallocca, Francesca De Nicola, Maurizio Fanciulli, Simona di Martino, Sara Bruschini, Christian Napoli, Alberto Ricci, Massimiliano Bassi, Federico Venuta, Erino Angelo Rendina, Gennaro Ciliberto, Rita Mancini

**Affiliations:** 1Department of Clinical and Molecular Medicine, Sant’Andrea Hospital, “Sapienza” University of Rome, 00161 Rome, Italy; claudia.devitis@uniroma1.it (C.D.V.); rita.mancini@uniroma1.it (R.M.); 2SAFU Laboratory, Department of Research, Advanced Diagnostic, and Technological Innovation, IRCCS “Regina Elena” National Cancer Institute, 00144 Rome, Italy; giacomo.corleone@ifo.gov.it (G.C.); matteo.pallocca@ifo.gov.it (M.P.); francesca.denicola@ifo.gov.it (F.D.N.); maurizio.fanciulli@ifo.gov.it (M.F.); 3Preclinical Models and New Therapeutic Agents Unit, IRCCS-Regina Elena National Cancer Institute, 00144 Rome, Italy; salvati.sv@gmail.com; 4Tumor Immunology and Immunotherapy Unit, Department of Research, Advanced Diagnostic and Technological Innovation, IRCCS Regina Elena National Cancer Institute, 00144 Rome, Italy; francesca.ascenzi@gmail.com; 5Pathology Unit, IRCSS “Regina Elena” National Cancer Institute, 00144 Rome, Italy; simona.dimartino@ifo.gov.it; 6Department of Experimental and Clinical Medicine, Magna Graecia University of Catanzaro, 88100 Catanzaro, Italy; sarabruschini@hotmail.it; 7Department of Medical Surgical Sciences and Translational Medicine, Sant’Andrea Hospital, “Sapienza” University of Rome, 00189 Rome, Italy; christian.napoli@uniroma1.it; 8Department of Clinical and Molecular Medicine, Division of Pneumology, Sapienza University of Rome, Sant’Andrea Hospital, 00189 Rome, Italy; alberto.ricci@uniroma1.it; 9Department of Thoracic Surgery, University of Rome Sapienza, 00161 Rome, Italy; massimiliano.bassi@uniroma1.it (M.B.); federico.venuta@uniroma1.it (F.V.); 10Department of Thoracic Surgery, Sant’Andrea Hospital, “Sapienza” University of Rome, 00189 Rome, Italy; 11Scientific Direction, IRCCS “Regina Elena” National Cancer Institute, 00144 Rome, Italy

**Keywords:** cancer stem cells, genome-wide, transcriptome, lung cancer, ATAC-seq, RNA-seq, CSCs, NSCLC, B4GALT1, LUAD

## Abstract

Background: According to the cancer stem cells (CSCs) hypothesis, a population of cancer cells with stem cell properties is responsible for tumor propagation, drug resistance, and disease recurrence. Study of the mechanisms responsible for lung CSCs propagation is expected to provide better understanding of cancer biology and new opportunities for therapy. Methods: The Lung Adenocarcinoma (LUAD) NCI-H460 cell line was grown either as 2D or as 3D cultures. Transcriptomic and genome-wide chromatin accessibility studies of 2D vs. 3D cultures were carried out using RNA-sequencing and Assay for Transposase Accessible Chromatin with high-throughput sequencing (ATAC-seq), respectively. Reverse transcription polymerase chain reaction (RT-PCR) was also carried out on RNA extracted from primary cultures derived from malignant pleural effusions to validate RNA-seq results. Results: RNA-seq and ATAC-seq data disentangled transcriptional and genome accessibility variability of 3D vs. 2D cultures in NCI-H460 cells. The examination of genomic landscape of genes upregulated in 3D vs. 2D cultures led to the identification of 2D cultures led to the identification of Beta-1,4-galactosyltranferase 1 (B4GALT1) as the top candidate. B4GALT1 as the top candidate. B4GALT1 was validated as a stemness factor, since its silencing caused strong inhibition of 3D spheroid formation. Conclusion: Combined transcriptomic and chromatin accessibility study of 3D vs. 2D LUAD cultures led to the identification of B4GALT1 as a new factor involved in the propagation and maintenance of LUAD CSCs.

## 1. Introduction

Lung cancer is the leading cause of cancer mortality worldwide [1]. Lung cancer can be divided into two major types, non-small cell lung cancer (NSCLC), which accounts for 80% of cases, and small cell lung cancer (SCLC). There are two main subtypes of NSCLC: Adenocarcinoma (LUAD) and squamous cell carcinoma (LUSC), of which LUAD is the most common [2,3]. The five-year survival rate of patients with advanced lung cancer remains very low, in spite of the introduction of targeted therapies for patients with specific genetic alterations such as EGFR mutations, or ALK and ROS translocations, and, more recently, of immunotherapy with anti-PD1 or anti-PD-L1 antibodies as first or second line therapy [4,5,6,7]. Unfortunately, failure of immunotherapy in NSCLC still occurs in a high proportion of cases because several factors, both tumor intrinsic and tumor extrinsic, are responsible for drug resistance [8,9]. In this scenario, lung cancer, and in particular its most frequent form LUAD, remains a highly unmet medical need in search of a deeper understanding of mechanisms of drug resistance and of the identification of new targets for more efficient therapy.

A widely accepted concept is that tumor cells are organized as a hierarchical population sustained by a subset of cells called Cancer Stem Cells (CSCs) capable of both symmetrical and asymmetrical divisions [10]. CSCs are resistant to conventional drugs and are responsible for tumor relapse after therapy. For these reasons, CSCs have been the object of intense study over the last two decades in the attempt to identify mechanisms responsible for their propagation and of ways to inhibit their growth.

Our group previously described a protocol able to propagate LUAD as 3D spheroids obtained either from malignant pleural effusions (MPEs) or from stable cell lines [11]. We showed that 3D spheroids are highly enriched in cells with stemness markers, including upregulation of Nanog, Oct4, SOX2, and ALDH1A1 activity [12,13]. Enrichment of CSCs was confirmed in transplantation studies in immune-deficient mice showing high rates of tumor engraftment [11,13]. Spheroids showed upregulation of the expression of a key protein in lipid metabolism, namely, Stearoyl–CoA–desaturase 1 (SCD1), the key factor in the biogenesis of monounsaturated fatty acids as precursors of phospholipids [14,15]. SCD1 pharmacological inhibition determined a decreased efficiency of 3D spheroid formation accompanied by a negative impact on the architecture of 3D spheroids. Moreover, SCD1 depletion induced a reduction in ALDH1A1 activity, a marker of cancer stem cells, determining apoptosis specifically in ALDH1A1-positive cells. Silencing of SCD1 impaired in vivo tumorigenicity of 3D lung cancer stem cells [11,16]. We also observed that LUAD 3D spheroids treated with an SCD1 inhibitor reverted resistance to cisplatin. Finally, SCD1 expression correlated with poor survival and worse clinical outcome [17]. Mechanistically, SCD1 inhibition induced the inactivation of β-catenin and YAP/TAZ signalling pathways [18]. Based on these observations and on those of several other laboratories, we consider the 3D spheroid assay a valuable approach to identify factors responsible for the maintenance and propagation of lung CSCs [19]. An intrinsic feature of CSCs is considered to be their plasticity, i.e., their ability to change their physiological properties, which has been linked to epigenetic reprogramming [20,21,22].

In this context, the study of epigenetic alterations in CSCs acquires particular relevance [23], also because epigenetic alterations might deregulate signalling pathways controlling self-renewal and differentiation, including Wnt, Myc, Notch, and Hedgehog pathways [24,25]. Many epigenetic studies on lung cancer focused on the methylation level of DNA [26,27]. Likewise, post-translational modifications of histone proteins are known to influence chromatin accessibility to transcription factors and complex gene expression machinery [28,29,30]. In recent years, ATAC-seq has been developed as an emerging technology able to provide a fast and sensitive analysis of the epigenome. ATAC-seq allows to obtain a chromatin accessibility map of the entire genome in order to identify functional gene-regulatory regions [31,32]. Hence, the intersection of transcriptomic analysis and of chromatin accessibility through ATAC-seq can provide a unique opportunity to better identify CSC-specific changes in gene regulatory networks.

In the present study, we have applied this approach to identify regulatory principles responsible for CSCs maintenance. For this purpose, we used the 3D spheroid assay on the MPEs derived LUAD cell line NCI-H460 and intersected transcriptomic with epigenetic signatures of CSCs-enriched (3D cultures) vs. differentiated cells (2D cultures) by RNA-seq and ATAC-seq, respectively. Interestingly, we observed that 3D cultures of NCI-H460 cells are characterized by the activation of pathways enriched of genes responsible for Epithelial to Mesenchymal Transition (EMT). Furthermore, we singled out and validated the role of the B4GALT1 as an emerging gene in CSCs propagation and as a target for their inhibition.

## 2. Materials and Methods

### 2.1. Cell Cultures

Established human NSCLC cell NCI-H460 was obtained from the American Type Culture Collection (ATCC, Manassasn, VA, USA) and was cultured according to the manufacturer’s instructions. BBIRE T-238 and BBIRE T-248 primary cultures were isolates from Malignant Pleural Effusion (MPEs) of patients with LUAD, as previously described [11,13]. The study was approved by the Ethics Committee (protocol number 1032/17). Cells in adherent condition were cultured in RPMI-1640 supplemented with 10% FBS, 1% L–Glutammine and 1% Penicillin/Streptomycin (Sigma, St. Louis, MO, USA), while 3D cultures in suspension were obtained as previously described [11,13,18]. Briefly, 20,000 cells/mL were resuspended in an appropriate amount of Spheroid Medium and seeded onto Ultralow Attachment plates (Corning Costar, MA, USA) to form 3D structures. After 4 days, the number of 3D spheres in single wells of a 96 well low attachment culture plate were counted, and their volume was estimated using Image J Software v1.50i. Cells were routinely checked for mycoplasma contamination by PCR.

### 2.2. RNA-Sequencing and Bioinformatic Analysis

Total RNA was extracted from NCI-H460 2D and 3D cells cultures, using Qiazol (Qiagen, Hilden, Germany), purified from DNA contamination through a DNase I (Qiagen) digestion step, and further enriched by Qiagen RNeasy columns profiling (Qiagen). Quantity and integrity of the extracted RNA were assessed by Nanodrop Spectrophotometer (Nanodrop Technologies LCC, Thermofisher, Waltham, MA, USA) and by Agilent 2100 Bioanalyzer (Agilent Technologies, Santa Clara, CA, USA), respectively. All RNA used for subsequent library preparation had an RNA integrity number greater than9.0. RNA libraries for sequencing were generated in triplicate using the same amount of RNA for each sample according to the standard Illumina TrueSeq Stranded Total RNA kit with an initial ribosomal depletion step using Ribo Zero Gold (Illumina, San Diego, CA, USA). The libraries were quantified by qPCR and sequenced in paired-end mode (2 × 75 bp) with NextSeq 500 (Illumina). Each sample was generated with the Illumina platform. A pre-step for quality control was performed to assess sequence data quality and to discard low quality reads [33]. Paired 75 bp long reads were quality checked and processed using Kallisto v0.46.0 [34] with parameters: “quant -t 8 -b 30” to the genome Ensemble GRCh38.96 [35] downloaded from https://github.com/pachterlab/kallisto-transcriptome-indices/releases/download/ensembl-96/homo_sapiens.tar.gz. Differential expression data were obtained running the R package Sleuth v0.27.3 [36] following the pipeline suggested in the Sleuth manual available https://pachterlab.github.io/sleuth_walkthroughs/trapnell/analysis.html using both available Likely ratio [37] and Wald tests [38]. Differentially expressed genes were considered those with FDR < 0.001 in both tests and B value over 0.3 for upregulation and less than −0.3 for downregulation. Volcano plots were drawn with an in-house R script (R v3.6.1). Gene set enrichment analysis was performed, submitting the top 100 significant genes to the hallmark.MSigDB [39] available in ShinyGo v0.60 [40] and plotted with an in-house R script.

### 2.3. Transcriptome Analyses of Public Datasets

TCGA cohort coupled with GTEx cohort was used to run a pan-cancer, LUAD, and LUSC specific analysis of the gene B4GALT1 expression. Box-plots and overall survival analysis were obtained using GEPIA v1 [41] (http://gepia.cancer-pku.cn/index.html) selecting default parameters. The Kaplan–Meier overall survival analysis of pooled B4GALT1 and SCD1 genes expression data was performed by re-analysis of the TCGA dataset available in Kaplan–Meier plotter [42,43] of 513 (LUAD) and 501 (LUSC) patients, respectively, with default parameters. Gene expression data were scaled in log_2_(TPM+1).

### 2.4. Assay for Transposase Accessible Chromatin with High-Throughput Sequencing (ATAC) and Bioinformatic Analysis

To profile open chromatin, we used the ATAC-seq protocol developed by Buenrostro et al., with minor modifications [44]. 50,000 NCI–H460 2D and 3D cells were washed once with 1x PBS and centrifuged at 500× *g* for 5 min at 4 °C. The cell pellet was lysed in ice-cold lysis buffer (10 mM Tris-HCl pH 7.4, 10 mM NaCl, 3 mM MgCl_2_, 0.1% IGEPAL CA–630) to isolate the nuclei. The nuclei were centrifuged at 500× *g* for 5 min at 4 °C and subsequently resuspended on ice in 50 μL transposase reaction buffer containing 2.5 μL of Tn5 transposase and 25 μL of 2xTD buffer (Nextera DNA Sample preparation kit from Illumina). After incubation at 37 °C for 30 min, the samples were purified with MiniElute PCR Purification Kit (Qiagen), eluting in 10 µL elution buffer (10 mM Tris–HCl pH 8). To amplify transposed DNA fragments, we used NEBNext High-Fidelity 2x PCR Master Mix (New England Labs, Ipswich, MA, USA) and the Customized Nextera PCR Primers. Libraries were purified by adding Agencourt Ampure XP (Beckman, Brea, CA, USA) magnetic beads (1:1 ratio) to remove remaining adapters (left side selection) and double purified (1:0.5 and 1:1.15 ratio) for right side selection. Libraries were controlled using a High Sensitivity DNA Kit on a Bioanalyzer. Each library was then paired-end sequenced (2 × 75 bp) on a NextSeq 500 instrument (Illumina). Paired-end 75 bp long reads were quality checked using FASTQC and aligned to Hg38 using Bowtie v 2.3.4.2 setting: “mode = local and p = 6”. The aligned reads were processed by Samtools v1.9 to be converted in BAM format then sorted and indexed. Peaks were called by MACS2 v2.1.2 with parameters “nomodel–shift–100–extsize 200–B–SPMR–call–summits”. Peaks with a –log10(q-value) lower than 2.0 were discarded. Peaks in bdg format were converted in bw format using bedGraphToBigWig v4 tool available on https://www.encodeproject.org/software/bedgraphtobigwig/. All peaks matching blacklisted regions were discarded from further processing. After peak generation, an in-house pipeline based on BEDTOOLS v2.25.0 and custom BASH scripts were developed to build a master list of accessible sites by pooling the significant peaks of all the sample dataset. The master list of accessible sites was produced with a multistep procedure: 1. To identify the common overlapping signal amongst all the samples, promoter peaks were intersected using BEDTOOLS multiinter. 2. The book-ended regions from the core signal file were merged using BEDTOOLS merge, then intersected with the original peak calls and sorted.

### 2.5. Comparative Analysis of DNA Accessibility

Differential analysis of the two groups (3D vs. 2D signals) was obtained using an in-house script which deploys edgeR suite (v3.28.0) [45]. Data were processed and normalized with the TMM method [46]. The differential comparison was performed using an in-house script which relies on the edgeR “exactTest” function (v3.28.0). Data were further adjusted for Benjamini Hochberg correction. The sites showing FDR < 0.05 were considered significant. Boxplot of differential enriched sites enrichment was performed with an in-house script. To show normalized read count differences observed between 3D and 2D culture, we developed an R script to build a heatmap showing differences between our groups. We centered the data to the row mean and fixed the color palette from the lowest to the highest value. Centered data were hierarchically clustered (Pearson distance, average linkage) using the hclust package. Results were visualized using heatmap.2 available in the gplot R package (v3.0.1.1).

### 2.6. Reverse Transcription Polymerase Chain Reaction (RT-PCR)

Total RNA was isolated by Trizol (Thermofisher) following the manufacturer’s instructions. First-strand cDNA was synthesized with PrimeScript RT reagent Kit, genomic DNA contamination is eliminated with gDNA Eraser (Takara Bio Inc, Kusatsu, Prefettura di Shiga, Japan). The cDNA was used for RT-PCR experiments carried out in a 7500 StepOnePlus (Applied Biosystems, Foster City, CA, USA) as previously described [47,48]. Primers for B4GALT1: 5′-CTATATCTCGCCCAAATGCTG-3′ (forward) and 5′-GTGCAATTCGGTCAAACCTC-3′ (reverse), and other primers are previously described [18]. The relative amount of all mRNAs was calculated using the comparative method (2^−∆∆Ct^) after normalization to H3. Three independent experimental replicates were performed.

### 2.7. siRNA Transfection

NCI-H460 cells were transfected with siRNA of B4GALT (s5726; Thermofisher) or control siRNA (Santa Cruz, CA, USA). DNA transfections were done with Lipofectamine RNAiMAX Reagents (Invitrogen Co, Carlsad, CA, USA) according to the manufacturer’s instructions.

### 2.8. Aldehyde Dehydrogenase (ALDH) Activity Assays

ALDH activity was performed with Activity kit (ab155893, Abcam, Cambridge, UK), in 3D transfected cells. Following the manufacturer’s recommendation, NADH standard and sample were added into a 96 well plate in duplicate. Subsequently, 50 µL of the Reaction mix, containing Acetaldehyde and ALDH substrate, were added to each well. Finally, it was incubated for 20–60 min at room temperature and measured at OD 450 nm. The activity was calculated according to the datasheet.

### 2.9. Statistical Analysis

All experiments were performed in triplicate and the statistical was carried out to GraphPad Prism v6.0 software. Group comparison were performed by Student’s test and values were expressed as mean ± Standard Error of Measurement (SEM).

## 3. Results

### 3.1. A Combinatorial Approach to Identify Gene Expression Features of Cancer Stem Cells (CSC)-Enriched Lung Adenocarcinoma (LUAD) Cell Cultures

In order to characterize regulatory cues responsible for orchestrating gene expression in CSCs, we used the approach described in Figure 1. The stable LUAD cell line NCI–H460 derived from MPEs was grown either in standard 2D attachment conditions or as 3D cultures enriched of CSCs markers [16,47] (Figure 1A). For transcriptomic analysis, RNA was extracted from both types of cultures. In parallel, in order to achieve genome-wide chromatin accessibility maps and to identify active gene regulatory regions, nuclei were isolated from both 2D and 3D cultures, chromatin was extracted and subjected to ATAC-seq (see materials and methods) (Figure 1B).

Transcriptomics technologies and chromatin accessibility assays are powerful tools to infer active cell regulatory states by analyzing the relative number of transcripts (Figure 1) and cis-regulatory activity, respectively. Although there are several difficulties in linearly correlating the activity of cis-regulatory regions with the relative production of gene transcripts, numerous studies [49,50,51] revealed that the large majority of regulatory elements are strongly linked to RNA production and occur in the proximity (within 50 kb) to the closest Transcriptional Start Sites (TSS). Thus, we took advantage of these observations and assayed the differences of RNA abundance and active chromatin accessibility sites in 3D vs. 2D cultures obtained from stable NCI–H460 cell line in a unique computational workflow. Each set of data was analyzed first individually then jointly to better identify gene regulatory elements capable of simultaneously controlling key cellular pathways specifically activated in CSC-enriched cell cultures.

### 3.2. Integrative RNA-seq and ATAC-seq Analysis Reveals Overarching Transcriptomic Features of Cancer Stem Cells (CSC)-Enriched LUAD Cell Cultures

RNA sequencing analysis revealed a global transcriptional rewiring from 2D to 3D cultures. The differential analysis revealed 3095 genes significantly enriched (FDR < 10^−3^) (detailed in Figure 2A, Appendix A) of which 1854 were downregulated and 1241 upregulated in 3D spheroids. To gain insights into the functional role of the differentially expressed transcripts, we computationally predicted the impact in their relative abundance to the cell physiological state (Figure 2B). Not unexpectedly, gene set enrichment analysis of the differentially regulated genes in the two systems revealed the emergence of a vastly different regulatory landscape. In particular, the 3D spheroids cell population exhibited a significant enrichment of gene pathways involved in oncogenesis and cancer progression, including EMT (FDR < 10^−13^), Hypoxia (FDR < 10^−10^), Cholesterol homeostasis (FDR < 10^−5^), and Apoptosis (FDR < 10^−4^) (Figure 2B).

Among the most significant upregulated genes, we identified 3 different isoforms of CD55 a well-known protein coding gene frequently linked to cancer aggressiveness in many carcinomas. Surprisingly, among the top 10% genes exhibiting the most significant transcript abundance shift (Appendix A), we found the gene B4GALT1. Notably, B4GALT1 protein has been previously associated with multi-drug-resistant cell phenotype in human leukemia [52] and other haematological malignancies [53,54]. On the other side, genes significantly downregulated were associated with MYC and E2F target genes (Appendix A). Importantly, our transcriptomic data further support the model of cancer aggressiveness recently proposed by Padmanaban et al. [55] in which significant loss of E–cadherin transcription (CDH1 gene) guides the upregulation of the transforming growth factor-β (TGFβ) expression (see Appendix A) as a requirement for metastatic invasion.

Then, ATAC-seq data were utilized to determine genome-wide chromatin accessibility of 3D vs. 2D NCI–H460 cells. DNA nucleosome-free regions store regulatory information in the form of active regulatory elements, including enhancers and promoters. These elements are highly plastic and act as essential players in guiding cell phenotypic states. The analysis revealed only 404 genomic loci (FDR < 0.05) significantly different between the two groups (see Figure 3A,B) of which 236 were closed and 168 open in 3D cultures, respectively.

Recent observations [56,57] confirmed the notion that the signal intensity of ATAC-seq is linearly dependent on the number of cells contributing to the signal. On the basis of this, we observed that the normalized signal of the open sites in 3D cells was consistently doubling the signal of sites only accessible in 2D cells, thus implying that 3D cultures are composed of a more homogeneous cell population compared to the 2D culture signal (Appendix A). Finally, we tested the relationship between gene expression and chromatin openness among each gene-specific neighborhood (|50 kb| to the TSS) to identify regulatory patterns exhibiting both significant changes in transcriptomic abundance and chromatin plasticity (Appendix A). By integrating RNA-seq and ATAC-seq data, we found that only a small proportion of differential ATAC-seq signal was occurring within |50 kb| from the TSSs of differentially regulated genes (Appendix A). However, the largest proportion of promoters associated with expressed genes were consistently accessible to regulatory factors in both groups, suggesting active transcription (Appendix A). These data imply that the differential abundance of transcripts in 3D vs. 2D cultures does not depend only on chromatin shifts in the vast majority of cases, but may be favored by other mechanisms whose further investigation could shed light on their identity (Appendix A). Examining the genomic landscape of the genes upregulated in 3D, we identified B4GALT1 as the top candidate. Interestingly, the B4GALT1 locus presents three regulatory sites (Figure 3C) differentially opened in 3D vs. 2D cell cultures. These regulatory sites may represent active enhancers; however, no previously published data annotate these genomic sites as regulatory regions, and further investigations need to be conducted to confirm this hypothesis.

In order to confirm B4GALT1 upregulation in LUAD 3D spheroids we carried out qRT-PCR assays on RNA extracted from 2D vs. 3D cultures of NCI–H460 cells and from two additional primary LUAD cell lines isolated from malignant pleural effusions in our laboratory (BBIRE–T238 and BB–IRE T–248). The results (Figure 3D) show that B4GALT1 is strongly upregulated in 3D cultures in all cases analyzed.

### 3.3. Validation of B4GALT1 as Novel Factor Responsible for the Propagation of Lung Adenocarcinoma (LUAD) Cancer Stem Cells (CSCs)

As shown in the previous paragraph, the combined RNA-seq/ATAC-seq analysis of 3D vs. 2D cultures of NCI–H460 cells highlighted B4GALT1 as one of the top genes overexpressed in CSCs enriched 3D cultures with a corresponding highly opened chromatin structure of its gene regulatory domains. In order to validate B4GALT1 as a critical factor for CSC propagation and a potential target for intervention, we carried out a series of analyses. We first performed an explorative analysis of B4GALT1 expression by interrogating TCGA and GTEx dataset. Remarkably, 84% of cancer types exhibit upregulation of B4GALT1 expression compared to healthy tissues, including both lung cancers LUAD and LUSC (see Figure 4A, Appendix A).

These data were consistent with Immunohistochemistry of normal and pathologic tissues available on the Human Protein Atlas data portal [58]. Consistent with these predictions, a meta-analysis of the TCGA datasets re-computed from raw RNA-seq by XENA project revealed that higher expression of B4GALT1 mRNA is linked with poor survival in many cancers (data not shown). In particular, LUAD patients with higher expression of B4GALT1 at diagnosis have a significantly worse outcome; conversely, this is not occurring in LUSC patients (Figure 4A). In agreement, a meta-analysis of pooled B4GALT1 and SCD1 mRNA expression data from TCGA conducted to similar results (Figure 4B), thus suggesting a potential co-regulatory activity in tumor progression of SCD1 and B4GALT1 genes.

Furthermore, in order to assess the role of B4GALT1 in the generation of 3D spheroids we transfected a B4GALT1 siRNA in NCI–H460 lung cancer cells. Transient knockdown of B4GALT1 strongly impaired 3D spheroids formation, both as to number and size (Figure 5A,B). The role of B4GALT1 in the propagation of lung CSCs enriched cell cultures was confirmed by the demonstration that its depletion reduced the expression level of other stemness markers such as Oct4, Sox2, and Nanog compared to scramble siRNA (Figure 5C). Moreover, the silencing of B4GALT1 affected SCD1 mRNA levels in 3D cells. To further confirm the role of B4GALT1 in CSCs, ALDH activity was also evaluated. As shown in Figure 5D, inhibition of B4GALT1 resulted in a significant reduction of ALDH activity in 3D cells, as well as a decrease of Nanog protein expression compared to control scrambled siRNA (data not shown). These results taken together suggest that B4GALT1 plays an important role in the propagation and maintenance of CSC-enriched LUAD cell cultures.

## 4. Discussion

In this study, in order to better identify transcriptional cues active in lung adenocarcinoma stem cells, we sought to link genes undergoing expression changes to their regulatory elements by leveraging RNA-seq and ATAC-seq data obtained from 2D and CSC-enriched 3D cell cultures. Indeed, the parallel profiling of gene expression and chromatin accessibility within the same cell bulk is a well-described approach to reveal causal regulatory relationships [59]. In this respect, our data revealed a global shift of gene expression which was accompanied by discrete changes in chromatin accessibility. Using a rigorous and conservative computational strategy, we identified more than 3000 differentially regulated transcripts and approximately 400 cis-regulatory regions affected in the passage from 2D to 3D cultures. In this regard, modifications of chromatin openness are linked to the activation and inhibition of regulatory pathways able to confer a selective growth advantage to cancer cells. Regulatory regions such as enhancers are key distal cis-regulatory elements that elevate the expression of nearby genes, independently from the distance to the target gene or orientation [60,61]. Our data predict that significant chromatin openness changes might collectively affect several hundreds of genes. Interestingly, both sets of differentially opened regulatory regions and RNA transcribed enrich for “cell migration” and “EMT pathways”, which are both distinctive features of cancer stem cells [62,63], and which we know from previous studies to be enriched in 3D vs. 2D cultures [64,65,66,67,68].

One of the most exciting findings of our combined transcriptomic and epigenomic analysis was the identification of B4GALT1 as one of the top candidates. Indeed, besides being one of the most highly transcriptionally upregulated genes in 3D cultures, the B4GALT1 locus shows three genomic regions located downstream the transcriptional start site whose chromatin is accessible only in 3D cultures as compared with 2D cultures.

We believe these are regulatory regions which may act as enhancers. Further studies are needed to confirm this hypothesis. An apparent limitation of our study is that we have applied our approach so far only to one LUAD cell line. However, we are confident that our observation about the involvement of B4GALT1 is of more general significance because of various reasons. Firstly, B4GALT1 transcriptional upregulation in 3D LUAD spheroids was confirmed by RT-PCR, not only in NCI–H460 but also in other LUAD primary cells, generated in our laboratory from malignant pleural effusions of LUAD patients. Secondly, our meta-analysis of TCGA datasets revealed that overexpression of B4GALT1 is linked to poorer survival in LUAD patients.

B4GALT1 has been reported before to facilitate cancer cell proliferation, invasiveness, and metastasis in several cancer types [69,70]. However, we provide the first evidence that this gene may be involved in the propagation of cancer stem cells, namely in lung cancer. In this respect the previous observation in hematologic cancers that B4GALT1 is responsible for drug resistance by regulating the expression of P-gp and MDR- associated protein [52,69] acquires particular relevance given the known ability of CSCs to be drug resistant [71,72].

We have observed that B4GALT1 silencing potently inhibits the 3D spheroid formation and impairs the expression of a set of stem cell markers. Hence, we believe that B4GALT1 is needed for lung CSCs propagation. At the moment the mechanism by which B4GALT1 activity facilitates lung CSCs propagation can only be the object of speculations. The enzyme catalyzes the transfer of galactose from UDP–Galactose to N-linked sugar chains of glycoproteins and is therefore important for the biosynthesis of glycoconjugates which may be required to facilitate the formation and maintenance of cell–cell contact and resistance to anoikis during the formation of 3D structures in non-adherent conditions. Further studies are required to investigate about this possibility.

Finally, B4GALT1 expression is somehow linked in LUAD to the expression of SCD1, which has been the object of intense studies by our group and other laboratories in recent years. TCGA data clearly show that co-overexpression of B4GALT1 and SCD1 is a negative prognostic factor. Furthermore, our silencing data demonstrate that inhibiting the expression of B4GALT1 strongly reduces SCD1 expression in LUAD 3D spheroids. We tend to believe that this is not a direct effect, but rather an indirect consequence of decreased survival of lung CSCs linked to the absent expression of B4GALT1.

## 5. Conclusions

In conclusion, our study proposes for the first time the involvement of B4GALT1 in lung CSCs maintenance and propagation. Therefore, this enzyme can become a new potential target of intervention. It will be important in the future to assess the effect of in vivo inhibition of B4GALT1 in LUAD tumor growth either alone or combined with inhibitors of other CSC-enriched targets, such as SCD1.

## Figures and Tables

**Figure 1 jcm-08-01928-f001:**
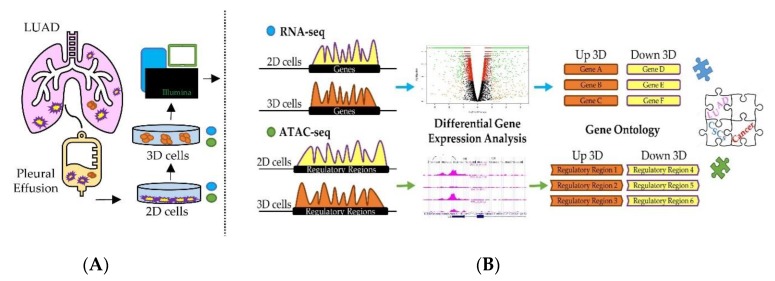
Graphical overview of the investigation. (**A**) 2D and 3D cell cultures were obtained from the stable NCI–H460 cell line obtained from the malignant pleural effusion of a lung adenocarcinoma (LUAD) patient. Total RNA was extracted and subjected to RNA-seq. Nuclei were isolated and processed to perform ATAC-seq. (**B**) Computational workflow developed to identify differentially expressed genes, pathways and biological processes in 3D compared to 2D cell cultures.

**Figure 2 jcm-08-01928-f002:**
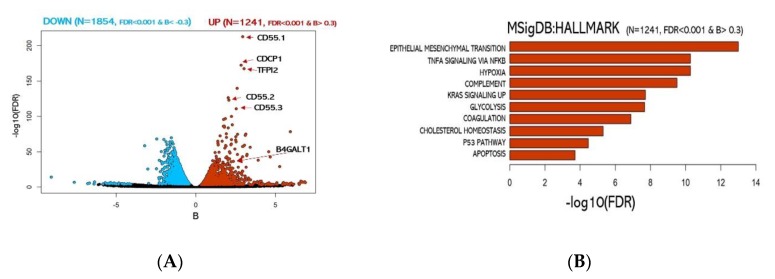
Analysis of 2D vs. 3D culture transcriptomes. (**A**) Volcano plot of over-expressed or under-expressed genes in 3D transcriptome vs. 2D. (blue: significantly under-expressed transcripts; red: significantly over-expressed transcripts). (**B**) Functional enrichment for upregulated genes in 3D cultures.

**Figure 3 jcm-08-01928-f003:**
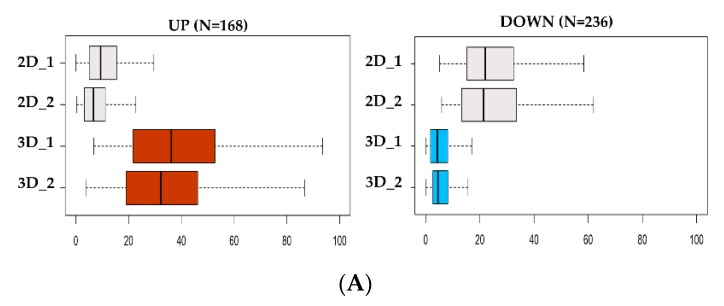
Analysis of DNA accessibility in 3D vs. 2D cultures. (**A**) ATAC-seq signal enrichment of normalized read counts at differential accessible sites in 3D vs. 2D (red = open; blue = closed). (**B**) Heatmap of differential chromatin accessibility sites (N = 404) showing the cell condition specificity of 2D and 3D ATAC-seq peaks. (**C**) Regulatory landscape of B4GALT1 locus. Peaks represent Assay for Transposase Accessible Chromatin with high-throughput sequencing (ATAC-seq) raw signals. (**D**) Reverse transcription polymerase chain reaction (RT-PCR) validation level of B4GALT1 upregulated in 3D cells compared to 2D culture in NCI-H460 (stable lung cell line) and BBIRE T-238, BBIRE T248 (primary lung cell lines). H3 reference gene have been used for normalization. Bars represent the mean of three biological replicates and technical replicates with their corresponding standard error of the mean (SEM). *** *p* < 0.0002; * *p* < 0.02; ** *p* < 0.0085.

**Figure 4 jcm-08-01928-f004:**
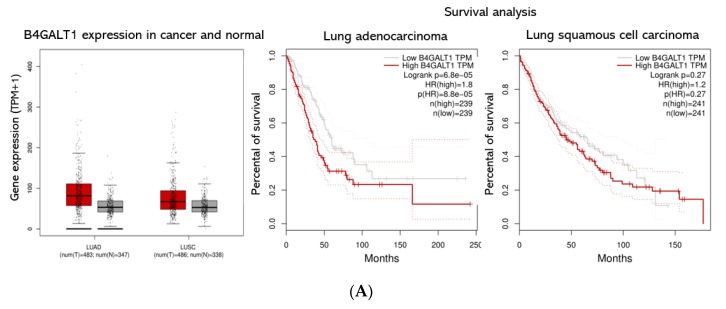
B4GALT1 gene expression and survival analysis in lung adenocarcinoma (LUAD) and lung squamous cell carcinoma (LUSC) patients. (**A**) On the left panel is shown a Box Plot of B4GALT1 expression in lung adenocarcinoma (LUAD) and lung squamous cell carcinoma (LUSC). Each dot represents a patient. (red: cancer tissues; grey: healthy tissues). On the right panel is shown a survival curves depict the B4GALT1 prognostic value in LUAD (N = 239 high expression tissues +239 low expression tissues) and LUSC cohort (N = 241 high expression +241 low expression). Comparison of survival curves was performed using a log-rank (Mantel–Cox) test. HR = Hazard ration. Dotted lines represent the 95% of Confidence Interval. (**B**) Kaplan–Meier curves depict the cumulative prognostic value of B4GALT1 and SCD1 gene expressions in LUAD (N = 250 high expression tissues + 254 low expression tissues) and LUSC (N = 153 high expression tissues + 342 low expression tissues) Abbreviations: LUAD, lung adenocarcinoma; LUSC lung squamous cell carcinoma; num, number; T, tumor; N, normal.

**Figure 5 jcm-08-01928-f005:**
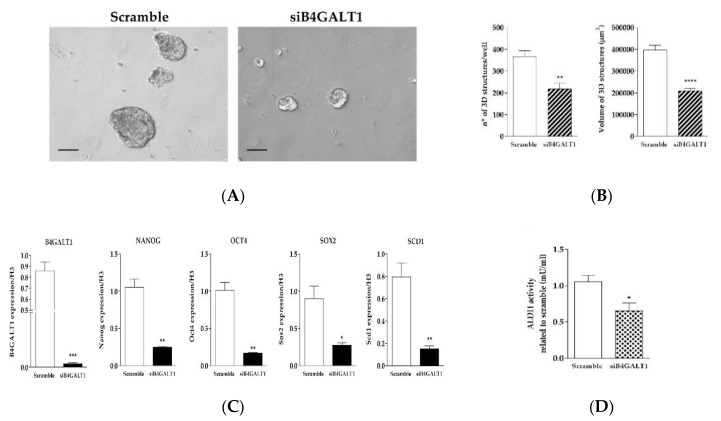
Knockdown of B4GALT1 RNA decreases the 3D structure formation and the expression of stemness markers. (**A**) Representative images of silencing of B4GALT1 reduction of 3D structure formation potential compared to the scramble of H460 cells. Scale bar = 100 µm. (**B**) Graphs show that silencing of B4GALT1 in 3D induces a decrease of volume and number of 3D spheroids. Number and volume of the 3D cells counted in each well after four days of culture. (**C**) Validation of B4GALT1 silencing in 3D cells, the results show a strongly decreases of stemness markers mRNA levels, such as Oct4, Sox2, Nanog, and SCD1. Expression of each gene was normalized to that of H3. (**D**) ALDH activity decrease substantially in 3D siB4GALT1 vs. Scramble cells. Experiments were performed in triplicate, and the background interference and the blank value were subtracted from the absorbance of the samples. In the bar plots, the mean ± standard error of mean (SEM) was shown from at least three independent experiments * *p*< 0.05, ***p*< 0.005, **** *p*< 0.0001 (vs. scramble).

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
