# Peer review of "B4GALT1 Is a New Candidate to Maintain the Stemness of Lung Cancer Stem Cells"

_jcm, 2019, doi:10.3390/jcm8111928_

Round 1

Reviewer 1 Report

This study of De Vitis and collaborators aims to identify molecules implicated in lung cancer stem cell maintenance. As a model, they used the human lung cancer cell line NCI-H460 cultured either in adherent culture conditions (2D) or in spheroid culture conditions (3D) that favor the acquisition of stemness properties. They combined both RNA-sequencing and Assay for Transposase Accessible Chromatine ATC-sequencing to unveil genes differentially expressed between these two cell culture conditions. This led to the identification of B4GALT1 gene being upregulated in NCI-H460 spheres which is supposed to be enriched  CSC-specific genes. Transient knockdown of B4GALT1 expression reduced sphere-forming potential of H460 cells and expression level of some stemness markers. Thus, the authors propose a role for B4GALT1 in lung CSC maintenance.

The present study of  De Vitis and collaborators  is of significance for the treatment of lung cancer. This is the first report of a potential role of B4GALT1 in lung cancer stemness. However their conclusions require additional validation. Since the implication of B4GALT1  in lung CSC is the major point of this study, it is necessary to provide more experimental evidence demonstrating that silencing B4GALT1 expression affects lung CSC. To demonstrate the role of B4GALT1 in CSC, two important experiments are needed:

- limiting-dilution tumor transplantation experiments : compare tumorigenicity between siScr and siB4GALT1-treated H460 cells as measured by tumor incidence, latency, and growth rate. Similarly to sphere-forming potential, inactivation of B4GALT1 should decrease the frequency of tumor-initiating cells in siB4GALT1-treated H460 cell population.

- compare drug resistance of siScr and siB4GALT1-treated H460 cells. Since CSC populations are more resistant to conventional cancer therapies than non-CSC populations, a decrease in the frequency of CSC in siB4GALT1-treated H460 cell population should impair the global cell resistance of the cell population to chemo- or radio-therapies.

Here, such experimental investigations are missing, without which the significance of the study is moderately felt. I would suggest the authors to perform these additional experiments to strengthen the connection between B4GALT1 and lung CSC.

Minor points :

- It has been shown previously that in some cases large spheres may actually contained fewer stem cells than smaller spheres. In order to evaluate the effect of siRNA treatment on sphere-forming potential of NCI-H460 cells, a secondary sphere formation assay must be performed.

- The title should be more informative about the function of B4GALT1 in cancer stem cells . « B4GALT1 as a new factor for CSC » is vague.

- page 3/17 line 114 : Cells in adherent conditions were cultured in RPMI-1640. No serum or antibiotics ?

- page 5/17 line 197 : verify the sequence of the reverse primer ; see NCBI reference sequence NM_001497.3.

- Most of the references are incomplete without volume and page numbers.

Author Response

REVIEWER 1

 Comments and Suggestions for Authors

This study of De Vitis and collaborators aims to identify molecules implicated in lung cancer stem cell maintenance. As a model, they used the human lung cancer cell line NCI-H460 cultured either in adherent culture conditions (2D) or in spheroid culture conditions (3D) that favor the acquisition of stemness properties. They combined both RNA-sequencing and Assay for Transposase Accessible Chromatine ATC-sequencing to unveil genes differentially expressed between these two cell culture conditions. This led to the identification of B4GALT1 gene being upregulated in NCI-H460 spheres which is supposed to be enriched CSC-specific genes. Transient knockdown of B4GALT1 expression reduced sphere-forming potential of H460 cells and expression level of some stemness markers. Thus, the authors propose a role for B4GALT1 in lung CSC maintenance.

We thank the reviewer for the positive evaluation of our manuscript. Here below we provide a point-by-point response to the reviewer comments.

The present study of De Vitis and collaborators is of significance for the treatment of lung cancer. This is the first report of a potential role of B4GALT1 in lung cancer stemness. However their conclusions require additional validation. Since the implication of B4GALT1 in lung CSC is the major point of this study, it is necessary to provide more experimental evidence demonstrating that silencing B4GALT1 expression affects lung CSC. To demonstrate the role of B4GALT1 in CSC, two important experiments are needed:

- limiting-dilution tumor transplantation experiments: compare tumorigenicity between siScr and siB4GALT1-treated H460 cells as measured by tumor incidence, latency, and growth rate. Similarly to sphere-forming potential, inactivation of B4GALT1 should decrease the frequency of tumor-initiating cells in siB4GALT1-treated H460 cell population.

We thank the reviewer for the suggestions. It will be important in the future to assess the effect of in vivo inhibition of B4GALT1 in LUAD tumor growth. Indeed, we plan to perform such experiments creating stable B4GALT1-silenced LUAD cell lines through shRNA vectors to evaluate the tumor incidence, latency and growth rate in mouse model.

- compare drug resistance of siScr and siB4GALT1-treated H460 cells. Since CSC populations are more resistant to conventional cancer therapies than non-CSC populations, a decrease in the frequency of CSC in siB4GALT1-treated H460 cell population should impair the global cell resistance of the cell population to chemo- or radio-therapies.

We thank the reviewer for the suggestions. We performed the treatment of Cisplatin in 3D cells scr vs 3D siB4GALT1. A preliminary data show a decrease of viable 3D siB4GALT1 cells treated with different concentration of Cisplatin compared to control (scramble). The treatment with CDDP was evaluated after 72h by cell titer glo. The number of viable lung cancer cells (NCI-H460) was mesured by quantification of the ATP present according to CellTiter-Glo® Luminescent Cell Viability assay. Unfortunatelly, due to the short time allowed for the revision of the manuscript we have been to complete either to carry out the triplicate biological or to add other cell lines. In the next study we will deepen this aspect.

Here, such experimental investigations are missing, without which the significance of the study is moderately felt. I would suggest the authors to perform these additional experiments to strengthen the connection between B4GALT1 and lung CSC.

Minor points:

- It has been shown previously that in some cases large spheres may actually contained fewer stem cells than smaller spheres. In order to evaluate the effect of siRNA treatment on sphere-forming potential of NCI-H460 cells, a secondary sphere formation assay must be performed.

The reviewer is right. We can’t carry out a secondary sphere formation assay to evaluate the effect of siRNA treatment because the silencing have a transient knowdown effect.

- The title should be more informative about the function of B4GALT1 in cancer stem cells. «B4GALT1 as a new factor for CSC» is vague.

We thank the reviewer for the suggestion. We change the title with:“ B4GALT1 is a new candidate to maintain the stemness of Lung Cancer Stem Cells”.

- page 3/17 line 114: Cells in adherent conditions were cultured in RPMI-1640. No serum or antibiotics?

We have added the medium supplemented with 10% FBS, 1% L-Glutammine and 1% Penicillin/streptomycin.

- page 5/17 line 197: verify the sequence of the reverse primer; see NCBI reference sequence NM_001497.3.

The reviewer is right. We have now corrected our previous mistake replacing the correct.

- Most of the references are incomplete without volume and page numbers.

The reviewer is right. We complete the reference with volume and page numbers.

Reviewer 2 Report

As authors revealed B4GALT1 is the candidate of factor to maintain the stemness in CSCs. Although it’s new finding and methodology is reasonable, the results are insufficient as follows.

Major points

The responsibility of B4GALT1 is thought to be highlight in this article. However, Figure 4 (a) – (c) are unclear and difficult to distinguish. Figure 4 is must to be enlarge or upregulated in resolution. Figure 4 (b) looks to have 6 lines. Authors should explain each line. The clinical outcome is important and polite explanation is needed.

The function of B4GALT1 in CSCs is also important. Although authors revealed the sphere formation assay and stemness marker expression in mRNA by B4GALT1 silencing. As authors know, the stemness markers expression is complicatedly regulated transcriptionally and post-transcriptionally. Moreover, the expression in mRNA level is not necessarily related to that in protein level. The confirmation of protein expression should be done. The result of two or more cell lines is preferable.

Author Response

REVIEWER 2

Comments and Suggestions for Authors #2

As authors revealed B4GALT1 is the candidate of factor to maintain the stemness in CSCs. Although it’s new finding and methodology is reasonable, the results are insufficient as follows.

Major points

The responsibility of B4GALT1 is thought to be highlight in this article. However, Figure 4 (a) – (c) are unclear and difficult to distinguish. Figure 4 is must to be enlarge or upregulated in resolution. Figure 4 (b) looks to have 6 lines. Authors should explain each line. The clinical outcome is important and polite explanation is needed.

We apologize for potentially misleading the reviewer on this point. Figures have been rebuilt at higher resolution and attached to the manuscript with a larger format. We thank the reviewer for raising this critical point and apologise for the confusion that Figure 4B might have generated. The Survival plot shows not only Cox proportional hazard ratio of High (red line) vs Low (grey line) B4GALT1 expression in LUAD and LUSC patients but also the relative 95% Confidence Interval information (high: red dotted line; low: grey dotted line). We have completely revised and corrected the figure caption.

The function of B4GALT1 in CSCs is also important. Although authors revealed the sphere formation assay and stemness marker expression in mRNA by B4GALT1 silencing. As authors know, the stemness markers expression is complicatedly regulated transcriptionally and post-transcriptionally. Moreover, the expression in mRNA level is not necessarily related to that in protein level. The confirmation of protein expression should be done. The result of two or more cell lines is preferable.

We thank the reviewer for the suggestions. We demostrated by western blotting that silencing of B4GALT1 affects Nanog protein expression compared to scrambled in 3D cell culture. Indeed, preliminary data show a slight downregulation of NANOG protein expression following siB4GALT1 in 3D cultures derived from NCI-H460 (stable cell line) and BBIRE-T248 (primary cell line). We strongly hope that western blot results have strenghtened this finding.

Round 2

Reviewer 1 Report

Concerning the minor points, the authors have addressed my comments in their revision to improve their manuscript. Concerning the major points, the authors have not provided any new data. However, I understand that the experiments that were suggested could not be performed in the short time allowed for the revision. The authors have then changed the title which is now more tempered concerning the potential role of B4GALT1 in lung CSC. Considering the significance of this first report on a potential role of B4GALT1 on lung cancer stem cells and its potential value as a target for more efficient therapies, the manuscript may be considered for acceptance. I urge the authors to rapidly perform the suggested experiments in order to deepen ad strengthen this initial report.

Reviewer 2 Report

I confirmed that Figure 4 was improved to distinguish.  I think it is enough to understand. Although Nanog expression is originally weak, the expression seems to be decreased bi silencing. I agree these improvements.